# Effectiveness of evidence based mental health apps on user health outcome: A systematic literature review

**Yeganeh Shahsavar, Avishek Choudhury**[iD]*

Industrial and Management Systems Engineering, Benjamin M. Statler College of Engineering and Mineral Resources, West Virginia University, Morgantown, West Virginia, United States of America

* avishek.choudhury@mail.wvu.edu

## Abstract

Research shows that over 70% of individuals globally who require mental health services lack access to adequate care. Mobile health (mHealth) technologies, such as phone apps, can be a potential solution to this issue by enabling broader and more affordable reach, thus addressing the problem of limited access to care. This study evaluates the effectiveness of evidence-based health apps on user mental health outcomes, particularly depression, anxiety, and suicidal behaviors. A comprehensive literature search was conducted using PubMed, Web of Science, and IEEE databases. In total, 6894 studies were identified, and 38 studies were selected for the review—thirty out of 38 studies employed randomized controlled trial designs. We identified 35 unique mobile apps. All the apps leveraged Cognitive Behavioral Therapy-based approaches. The most common approaches were context engagement and cognitive change, highlighting a significant focus on using personalized engagement activities and empowering users to alter their perspectives and reframe negative thoughts to improve their mental health. While mental health apps generally positively impact mental health outcomes, the findings also highlight significant variability in their effectiveness. Future studies should prioritize long-term effectiveness, wider reach to ensure it suits a diverse range of people, and the employment of objective evaluation methodologies.

## Introduction

Mental health issues have become a significant global public health concern [1]. The World Health Organization (WHO) has projected that by 2030, mental illness will become the primary global disease burden [2]. A systematic review indicated that approximately 14.3% of deaths worldwide, equivalent to around eight million deaths annually, are linked to mental disorders [3]. In the European Union region, an estimated 165 million individuals are affected by mental illnesses each year, predominantly anxiety, mood, and substance abuse disorders [4]. Additionally, a study conducted in Ethiopia revealed that individuals with severe mental health problems have life expectancies 30 years shorter than those without such conditions [5]. Globally, nearly one million people die by suicide annually, with three-quarters of individuals with mental health problems residing in low- and middle-income countries, where less

**Data availability statement:** All relevant data are within the paper and its Supporting information files.

**Funding:** This work was supported by the West Virginia University Internal Seed Grant (#3086). "The funders had no role in study design, data collection and analysis, decision to publish, or manuscript preparation.

**Competing interests:** The authors have declared that no competing interests exist.

than one in ten receive evidence-based treatment [5]. In the USA, suicide ranks as the second leading cause of death among university students [6]. Additionally, it is estimated that by 2023, mental health-induced reduction in productivity will cost the global economy approximately $16 trillion [7]. While precise global figures on the prevalence of mental health problems may vary, the available data indicate a significant burden of mental disorders worldwide.

Research shows that over 70% of individuals globally who require mental health services lack access to adequate care, contributing to a widening mental health treatment gap [8]. Various barriers hinder individuals from receiving optimal mental health care [9]. System-level barriers, such as difficulties in detecting mental health concerns, limited availability of services, inconsistent pathways to care, and affordability issues, continue to impede access to mental health services [10–13]. Furthermore, stigma, lack of awareness, sociocultural factors, and geographical inaccessibility act as significant barriers that prevent individuals from utilizing mental health services [14,15]. Efforts to enhance access to care have been explored by implementing collaborative care models and integrated service delivery approaches [16,17]. However, challenges persist, such as limited resources in rural areas, disparities in resource distribution, and inadequate support for vulnerable populations [18,19]. The COVID-19 pandemic has further highlighted the inadequacies in mental health care accessibility, leading to a global mental health crisis [20–22].

Efforts to enhance global mental health include increasing access to mental health services [23]. Mobile health (mHealth) technologies, such as phone apps, can be a potential solution to this issue by enabling broader and more affordable reach, thus addressing the problem of limited access to care [24–26]. The rapid increase in the use of mobile phone applications has created an opportunity to enhance access to evidence-based care [27]. In 2018, approximately 325,000 mobile health apps were available, with about 200 being launched daily [28]. mHealth apps have provided new avenues to reach populations that were previously challenging to access through traditional healthcare channels [29]. The scalability of app-based interventions has been suggested as a strategy to tackle the global burden of mental illnesses and offer services to individuals who may have had limited access to care [30]. Studies have indicated that mobile apps can effectively screen for mental health conditions, such as depression, and encourage users with high depressive symptoms to seek help from healthcare professionals [31].

However, there is a lack of understanding about the types of mHealth apps that are effective beyond screening. Existing research highlights that most publicly available mental health apps are not evidence-based and may even pose risks to users [26,32]. A 2022 study points out methodological issues and a lack of robust evidence regarding the effectiveness of these apps in changing behaviors or improving clinical outcomes [33]. Few existing reviews in this field have focused on the feasibility of various apps designed for agoraphobia, eating disorders, post-traumatic stress disorders, substance use disorders, and sleep disorders [34,35]. However, there is a lack of evidence reviewing the effectiveness of mobile apps in improving mental health outcomes [36,37].

In our review, we focused on the effectiveness of evidence-based apps designed using both randomized and non-randomized controlled trials to influence user mental health outcomes, particularly depression, anxiety, and suicidal behaviors. Our review discusses the methodology leveraged by such apps to improve mental health.

## Methods

This systematic review is reported according to the Preferred Reporting Items for Systematic Reviews and Meta-Analysis (PRISMA) guidelines (see S1 File) [38]. The detailed protocol (osf. io/x6m7u) is registered at the Open Science Framework [39].

## Search strategy

The search strategy was developed based on People, Intervention, Comparison, and Outcome (PICO) criteria. The population was participants using a mobile app designed to reduce mental health problems; the outcome was the impact of the app on user mental health; the intervention was a mobile app; the comparison was made by classifying the interventions based on their functions and user health outcome. A comprehensive literature search was conducted in PubMed, Web of Science, and IEEE Xplore for relevant articles. The search query consisted of the following: ((Depression OR anxiety OR suicid*) AND (Mobile OR app OR smartphone)) NOT (sleep OR alcohol OR drugs OR addiction OR tobacco) (see S2 File).

## Inclusion and exclusion criteria

Studies published in English and within the last ten years (January 2013 to September 2023) were included. We only included peer-reviewed clinical trial study designs that used mobile mental health apps to improve mental health outcomes. We focused on apps designed to address depression, anxiety, and suicidal behavior. Any article that did not assess the impact of an app on user mental health outcomes and solely focused on screening, app feasibility, or app development was eliminated. We also excluded apps that facilitated telehealth by connecting users with a clinician.

The methodology for selecting studies involved a multi-step process. Two authors independently selected the studies using the inclusion and exclusion criteria. Conflicts were then resolved with discussion, without the involvement of third parties. First, duplicates were identified and removed using Excel sheets created from the database exports. Titles of potentially relevant studies were then screened manually to eliminate irrelevant articles. A review of abstracts followed this to exclude studies that did not use mobile phone technology, did not focus on mental health apps, lacked emphasis on treatment or mental health impact, or addressed unrelated topics like mobile or technology addiction. Finally, full-text studies were evaluated against the inclusion and exclusion criteria.

## Data collection

For each article included in the final review, we recorded their objective, study design, participant age, survey instrument used by the study, mobile app name, its function, and the study's primary outcome. These data were extracted at face value, as reported in the reviewed articles. The app functions were then mapped to the principles of Cognitive Behavioral Therapy (CBT), mainly (a) context engagement, (b) attention change, and (c) cognitive change [40]. Context engagement focuses on helping people develop healthier associative learning patterns. Individuals are taught to recognize and respond to cues for threats and rewards in a more balanced and realistic way, leading to improved functioning. Attention changes technique aims to train individuals to direct their attention toward relevant, non-distressing stimuli. It includes therapeutic practices such as attention training, acceptance or tolerance training, and mindfulness. Cognitive change involves helping individuals shift their perspective on events to alter the emotional significance they attach to those events [41]. Methods like cognitive reframing and decentering are commonly used.

## Quality assessment and risk of bias

We conducted a quality assessment of the papers following the Mixed Methods Appraisal Tool (MMAT) (see S3 File) [42]. The MMAT assesses the quality of qualitative, quantitative, and mixed methods studies. It focuses on methodological criteria and includes the nature of

the study (randomized or nonrandomized clinical trial) across the performance, detection, attrition, and selection biases. No articles were excluded.

## Results

### Study selection and characteristics

Of the 9,547 abstracts initially identified, 2642 duplicates were removed. Of the remaining 6894 articles, 508 review articles were eliminated. We also eliminated 5982 of the remaining articles based on title screening. These articles were not at all related to our topic of interest. From the remaining 404 articles, 351 were eliminated after abstract screening (studies with no mobile app = 114; studies not about suicide, depression, or anxiety = 47; studies focused only on mental health assessment = 82; studies about phone addiction = 23; telehealth apps = 36; studies focusing only on usability = 49). The remaining 53 full texts were screened, of which 15 were eliminated (telehealth app = 6; no mobile app used = 2; no treatment outcome reported = 7). Fig 1 shows the final 38 full-text articles that met the inclusion criteria for the current systematic review (see S4 File).

Table 1 summarizes the objectives, study design, participants, and survey instruments. We identified 35 unique mobile apps across 38 studies. Thirty-five studies (32 apps) report significant improvements in depressive symptoms [43–66], anxiety [43–45,50–52,54,55,58,60–62,67–75], and suicidal behavior [52,76,77]. Thirty studies employed randomized controlled trial (RCT) designs [43–49,54–60,62–73,75–78], while the other eight clinical trial studies used a non-randomized controlled design [50–53,61,74,79,80].

### Mobile health apps characteristics

Table 2 introduces all the apps identified in the review and summarizes the CBT functions they use. We identified 3 CBT approaches: Context engagement, attention change, and cognitive change being used by the studies. Cognitive change was the most commonly used approach, implemented in 30 apps, followed by context engagement in 24 apps and attention change in 23 apps.

### Context engagement

We identified 24 apps that used context engagement methods like video playback, motivational words, guided self-assessments, physical activity tracking, daily health tips, and gamified challenges. Using context engagement approach, 17 apps were effective in reducing depression symptom: Smartphone Positive Stimuli Response System (SPSRS) [44,53], indoor exercise (IE) app [43], the ibobbly app [46], the Problem-Solving Therapy (iPST) app [47], Mood Mission app [51,67], Blue Ice app [52], Headspace [48,49], Smiling Mind [48], Mello App [54], Coping Camp app [55], Down Dog app [56], eQuoo app [60], MindLAMP app [61], FertiStrong app [62], Welzen app [63], We'll App [65], MoodHacker app [66].

The following 15 apps also reduced anxiety: Smartphone Positive Stimuli Response System (SPSRS) [44], indoor exercise (IE) app [43], Mood Mission app [51], Blue Ice app [52], Insight Timer app [68], Mello App [54], Coping Camp app [55], COVID Coach [70], ImExposure (IE) app [71], Headspace [72,73], eQuoo app [60], MindLAMP app [61], FertiStrong app [62], Aware App [74], Flowy [75].

Furthermore, the Therapeutic Evaluative Conditioning (TEC) app [76] was found to reduce suicidal plans and behaviors, the Loving-Kindness Meditation (LKM) app [77] was found to reduce suicidal ideation, and Blue Ice app [52] users reported a reduction in self-harm symptoms.

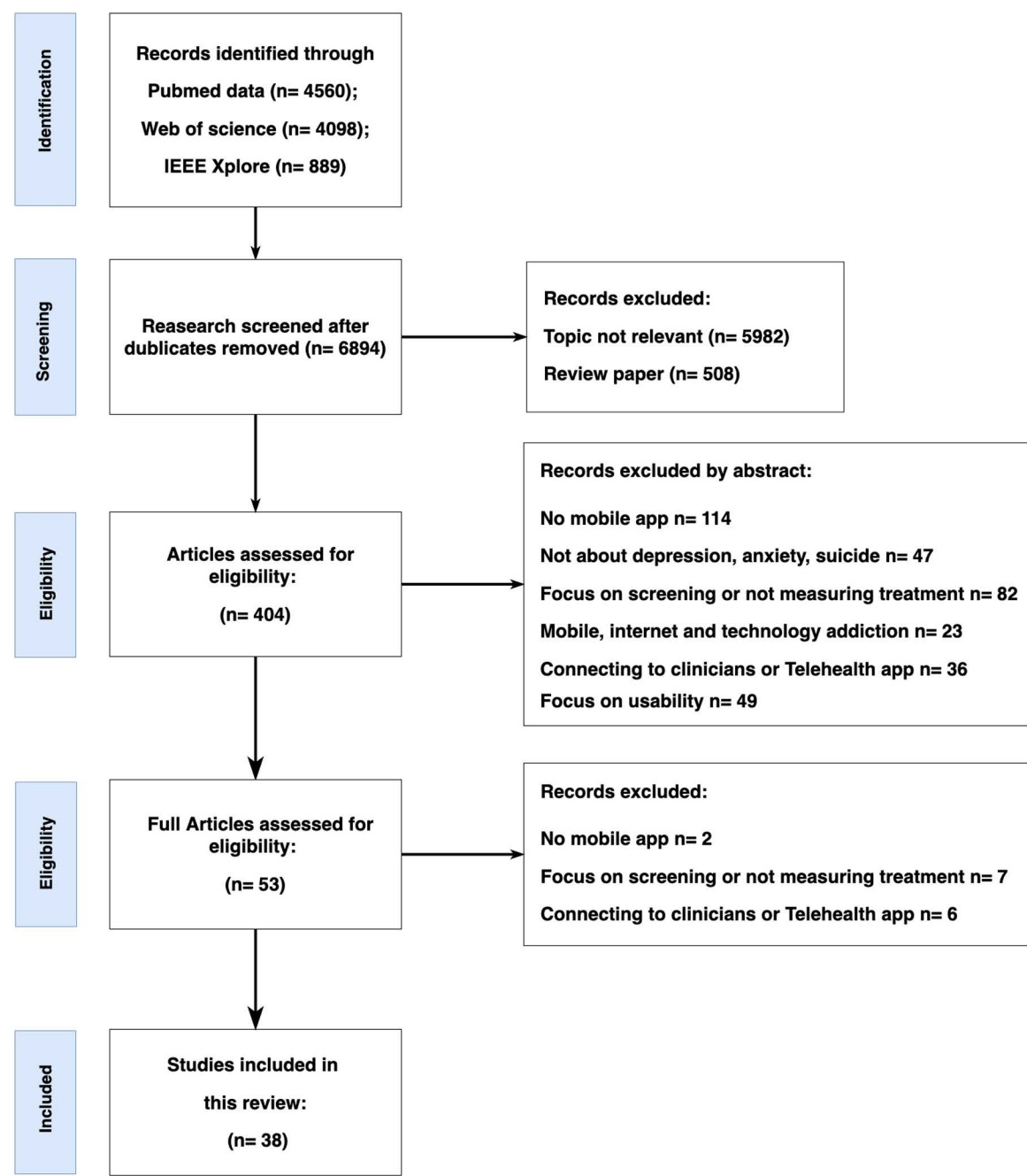

**Fig 1. The flow diagram detailing the review process and results at each stage.**

## Attention change

Twenty-three apps used attention change techniques such as mindfulness, acceptance, self-soothing techniques, and attention bias modification training to improve mental health. The ibobbly app [46], Feel Stress Free app [45], MoodMission app [51,57], BlueIce app [52], Headspace [48,49] and Smiling Mind [48], Mello App [54], Coping Camp app [55], Down Dog app [56], Anchored app [58], Subliminal Priming with Supraliminal Reward Stimulation

**Table 1. The evidentiary table summarizes the primary objectives, app names, participant characteristics, study design and durations, and mental health outcomes (n = 38).**

| Study | Objective | App name | Participants | Study design | Study duration | Outcome |
|---|---|---|---|---|---|---|
| [53] | To improve self-confidence and depressive symptoms in people with subthreshold depression. | Smartphone Positive Stimuli Response System application | N = 22, Age = 18 to 24 | Longitudinal; Non RCT | 5 weeks | The Epidemiologic Studies Depression scores improved from 19.18 to 14.50. |
| [44] | To assess the impact of the Smartphone Positive Stimuli Response System app on people with subthreshold depression. | Smartphone Positive Stimuli Response System application | N = 32, Age = 18 to 24, Mean age = 20.06 (SD 1.24) | Longitudinal; RCT | 5 weeks | Significant improvement is noted in depression and anxiety scores. |
| [51] | To examine the connections between a mobile app with CBT strategies and mental health outcomes. | MoodMission app | N = 617, Age = 13 to 70, Mean age = 26.9 (SD 10.9) | Longitudinal; Non RCT | 30 days | The app engagement enhanced mental wellbeing and reduced depression and anxiety via coping self-efficacy in those with moderate baseline symptoms. |
| [57] | To investigate the efficacy of using the mental health application alongside treatment as usual in reducing depressive and anxiety symptoms among psychiatric outpatients. | MoodMission app | N = 48, Age = Control group (median = 26), intervention group (median = 27) | Longitudinal; RCT | 4 weeks | The app significantly reduced depressive symptoms but showed no significant improvement in anxiety symptoms. |
| [48] | To determine whether mindfulness meditation apps lead to improvements in mental health among university students. | Headspace app and Smiling Mind app | N = 208, Age = 18 to 49 | Longitudinal; RCT | 30 days | Both apps reduced depressive symptoms. Headspace improved mindfulness and maintained benefits with continued use. Smiling Mind enhanced resilience and adjustment. |
| [49] | To examine the effectiveness of a mindfulness meditation regimen incorporating gamification principles in reducing symptoms of depression among college students. | Headspace app | N = 72, Age>=18 | Longitudinal; RCT | 14 days | The app significantly reduced the severity of depression symptoms. |
| [72] | To determine the efficacy of a mindfulness-based phone application in reducing stress, anxiety, and burnout among orthopaedic surgery residents. | Headspace app | N = 24, Age = control group (mean 30.6, SD 2.8), intervention group (mean 31.0, SD 2.8) | Longitudinal; RCT | 8 weeks | The app significantly reduced stress, anxiety, emotional exhaustion, and depersonalization scores after using 8 minutes per day, 2 days per week in orthopaedic surgery residents. |
| [73] | To evaluate the effectiveness of short mobile meditation sessions in reducing anxiety and improving overall wellness among surgical residents and faculty members. | Headspace app | N = 19, Age = control group (mean 34.20, SD 4.61), intervention group (mean 41.11, SD 14.55) | Longitudinal; RCT | 14 days | The app significantly reduced anxiety levels but showed no significant improvements in depressive symptoms or professional quality of life. |
| [79] | To assess the impact of the mobile app on university students' mental health. | Mental App | N = 57, Age = 18 to 24 | Longitudinal; Non RCT | 2 weeks | There were no significant differences between control and intervention group. |
| [45] | To evaluate the effectiveness of a self-guided mobile app for the treatment of depression and anxiety symptoms in students. | Feel Stress Free app | N = 168, Age>=18 Mean age = 24.3 (SD 6.71) | Longitudinal; RCT | 6 weeks | The app significantly improved depression. Improvement in anxiety was weakly noted. |
| [76] | To develop an effective and scalable treatment for self-injurious thoughts and behavior that can be easily delivered on a large scale. | Therapeutic Evaluative Conditioning app | N = 114, 131, and 163, Age>=18, | Longitudinal; RCT | 4 weeks | The app reduced self-cutting episodes, suicide plans, and suicidal behaviors. The app had no impact on suicide ideation. |

*(Continued)*

**Table 1.** (Continued)

| Study | Objective | App name | Participants | Study design | Study duration | Outcome |
|-------|-----------|----------|--------------|--------------|----------------|---------|
| [50] | To examine the impact of a self-monitoring mobile app and the relationships between app engagement and mental health outcomes. | MoodPrism app | N = 198, Age = 13 to 69 | Longitudinal; Non RCT | 4 weeks | The app reduced depression and anxiety. |
| [80] | To evaluate an app that uses metacognitive questions to address mental health problems. | MindSurf app | N = 23, Age>=18 | Longitudinal; Non RCT | 2 weeks | No significant reductions in depression and anxiety were noted. |
| [67] | To impact on the stress, anxiety, and depression levels in the working population. | Interpretation bias evaluation app | N = 92, Mean age = 28 to 31 | Longitudinal; RCT | 12 weeks | Significant improvements in anxiety and stress were noted, but no improvement in depression. |
| [43] | To measure the effect of interpretation bias modification training app on the stress, anxiety, and depression of working population. | Indoor exercise app | N = 180, Age>=20 | Longitudinal; RCT | 10 weeks | The app improved depression and anxiety. |
| [78] | To test the hypothesis that a single session of app, lead to reductions in stress reactivity and anxiety. | Attention Bias Modification Training app | N = 42, Age = 18 to 38, | Cross-sectional; RCT | 2 hours | No significant reductions in depression and anxiety were noted |
| [46] | To assess the efficacy of the self-help mobile app in addressing suicidal ideation, depression, psychological distress, and impulsivity among Indigenous youth in remote Australian communities. | ibobbly app | N = 61, Age = 18 to 35, Mean age = 27.48 (SD 9.54) | Longitudinal; RCT | 6 weeks | Significant reductions in distress and depression. No substantial impact on suicidal ideation or impulsivity was noted. |
| [47] | To compare usage patterns and clinical outcomes for three different self-guided mobile apps designed based on cognitive training, treatment control, and evidence-based psychotherapy for depression. | Problem-Solving Therapy app, Cognitive Control app | N = 626, Age>=18 Mean age = 33.95 (SD 11.84) | Longitudinal; RCT | 12 weeks | Participants with higher baseline depression scores experienced better mood improvements with cognitive training and problem-solving apps. |
| [52] | To assess the effectiveness of a self-help app in assisting students with managing self-harm, and wellbeing. | BlueIce app | Baseline N = 80, Age = 17 to 52 (mean 21.17, SD 4.39); Follow-up N = 27, Age = 18 to 32 (mean 21.19, SD 2.82). | Longitudinal; Non RCT | 6 weeks | The app reduced self-harm, anxiety, and depression while increasing coping self-efficacy among university students. |
| [68] | To examine the effect of daily meditation app usage on adult anxiety and mental well-being during the COVID-19 pandemic. | Insight Timer app | N = 100, Age>=18 | Longitudinal; RCT | 30 days | The app reduced anxiety and improved well-being more effectively than the control group. |
| [54] | To examine the effectiveness of a mindfulness app targeting repetitive negative thinking in young people with depression and anxiety. | Mello app | N = 30, Age = 16 to 25 | Longitudinal; RCT | 6 weeks | The app significantly reduced depression, anxiety, and repetitive negative thinking. |
| [55] | To evaluate the effectiveness of the relaxation app in reducing stress, depression, and anxiety while improving stress-coping behaviors and mental health well-being among high school students. | Coping Camp app | N = 540, Age<= 18 | Longitudinal; RCT | 11 weeks | The app significantly reduced perceived stress, anxiety, and depression levels but did not notably enhance stress-coping behaviors or overall mental health well-being. |
| [69] | To examine the effectiveness of digital cognitive education app in reducing anxiety scores compared to guided brief cognitive behavioral therapy. | GAMA-AIMS | N = 66, Age (mean 20.45, SD 0.71) | Longitudinal; RCT | 8 weeks | The app significantly reduced anxiety scores, with a decrease observed from week 2 to week 8. |

*(Continued)*

**Table 1.** (Continued)

| Study | Objective | App name | Participants | Study design | Study duration | Outcome |
|---|---|---|---|---|---|---|
| [70] | To evaluate the efficacy of a mindfulness app in improving mental health outcomes among healthcare workers during the COVID-19 pandemic. | COVID Coach | N = 30, Age>=18 Mean age = 33.4 | Longitudinal; RCT | 4 weeks | Significant improvements in anxiety and acute stress disorder severity from pre- to post-intervention. |
| [56] | To assess the effectiveness of a 12-week app-based exercise intervention in reducing depressive symptoms, burnout, and absenteeism among healthcare workers. | Down Dog app | N = 288, Age>=18 (mean 41.0, SD 10.8). | Longitudinal; RCT | 12 weeks | The app significantly reduced depressive symptoms, cynicism, and emotional exhaustion. |
| [77] | To investigate the effectiveness of a short video app-guided loving-kindness meditation in enhancing positive psychological capital while reducing suicide ideation among college students. | Loving-Kindness Meditation app | N = 74, Age (mean 17.73, SD 1.59) | Longitudinal; RCT | 8 weeks | The app significantly increased self-compassion and positive psychological capital while reducing suicide ideation. |
| [58] | To assess the effectiveness of the mindfulness app in preventing depression onset and enhancing related outcomes among workers with moderate or higher stress levels. | Anchored app | N = 1084, Age = control group (mean 42.84, SD 10.20), intervention group (mean 43.08, SD 9.94) | Longitudinal; RCT | 4 weeks | The app significantly reduced depressive and anxiety symptoms, improved work performance, and prevented depression in one month and in highly engaged users after 6 months. |
| [71] | To evaluate the effectiveness of the imaginal exposure intervention in reducing social anxiety and increasing self-efficacy among individuals with social anxiety disorder. | ImExposure app | N = 82, Age>=18 | Longitudinal; RCT | 1 week | The app significantly reduced social anxiety and increased self-efficacy from pre- to post-treatment. |
| [59] | To evaluate the preliminary efficacy of the subliminal priming with supraliminal reward stimulation app in improving depressive mood immediately after a 10-minute video intervention in individuals with subthreshold depression. | Subliminal Priming with Supraliminal Reward Stimulation | N = 32, Age = 20 to 27 (mean 20.88, SD 0.72) | Cross-sectional; RCT | 10 minutes (Cross sectional intervention) | The app showed a small improvement in depressive mood. |
| [60] | To evaluate the effects of the mental health game app on resilience, anxiety, depression, and attrition in a student population. | eQuoo app | N = 1163, Age>=18 | Longitudinal; RCT | 5 weeks | The app significantly reduced anxiety and depression scores. |
| [61] | To investigate the preliminary effect size of the mindfulness app on reducing anxiety and depression symptoms | MindLAMP app | N = 484, Age = control group (mean 21.5, SD 3.9), intervention group (mean 35.4, SD 12.5) | Longitudinal; Non RCT | 28 days | The app showed small effects on improving anxiety and depression outcomes. |
| [62] | To evaluate whether a mobile application incorporating cognitive-behavioral techniques and relaxation strategies can reduce psychological distress in men facing infertility. | FertiStrong app | N = 38, Age = 25 to 48 (mean 33.7, SD 4.5) | Longitudinal; RCT | 4 weeks | The app showed a small decrease in anxiety and depression scores and significantly reduced infertility-related stress. |
| [63] | To evaluate the effectiveness of a mindfulness app in reducing depression, anxiety, and stress among university students and staff. | Welzen app | N = 561, Age>=18 | Longitudinal; RCT | 28 days | The app showed small but significant reductions in stress and depression, with no additional improvements observed in subjects who practiced more. |
| [64] | To evaluate the preliminary effectiveness of the cognitive behavioral therapy app, in reducing depressive symptoms among mothers during the very early postpartum period. | CareMom app | N = 112, Age (mean 31.9, SD 3.62) | Longitudinal; RCT | 4 weeks | The app significantly reduced depressive symptoms, with no significant changes observed in anxiety symptoms. |

*(Continued)*

**Table 1.** (Continued)

| Study | Objective | App name | Participants | Study design | Study duration | Outcome |
|---|---|---|---|---|---|---|
| [65] | To investigate the effects of mindfulness and social support theory app on parenting self-efficacy and postpartum depression symptoms in puerperae. | We'll App | N = 130, Age = 25 to 40 (mean 31.81, SD 5.36) | Longitudinal; RCT | 8 weeks | The app significantly increased perceived social support and maternal parental self-efficacy while significantly reducing postpartum depressive symptoms. |
| [74] | To assess the impact of a mindfulness meditation mobile app in reducing the perceived stress and anxiety. | Aware App | Cross-sectional: N = 222 (111 meditators, 111 non-meditators); Longitudinal: N = 67, Age>=18 | Cross-sectional and Longitudinal; Non RCT | Cross-sectional: After 90 days for meditators; Longitudinal: 21 days | The cross-sectional study showed significantly lower stress and anxiety levels in meditators compared to non-meditators, and the longitudinal study confirmed a substantial reduction in stress and anxiety. |
| [75] | To evaluate the clinical efficacy of the mobile health game app in managing anxiety, panic, and hyperventilation symptoms associated with chronic common mental health disorders. | Flowy | N = 63, Age>=18 | Longitudinal; RCT | 4 weeks | The game app showed improvement in anxiety outcome. |
| [66] | To evaluate the effectiveness of the mobile web app in reducing depression symptoms, negative thoughts, and workplace distress, among employed adults with mild-to-moderate depression. | MoodHacker app | N = 300, Age = alternative care group (mean 40.6 SD 11.5), intervention group (mean 40.7, SD 11.2) | Longitudinal; RCT | 6 weeks | The app significantly reduced depression symptoms, negative thoughts, and workplace distress. |

(SPSRS) [59], eQuoo app [60], MindLAMP app [61], Welzen app [63], We'll App [65], and MoodHacker app [66] showed a positive effect in reducing depression.

MoodMission app [51], BlueIce app [52], Feel Stress Free app [45], Insight Timer app [68] Mello App [54], Coping Camp app [55], GAMA-AIMS [69], COVID Coach [70], Anchored app [58], ImExposure (IE) app [71], Headspace [72,73], eQuoo app [60], MindLAMP app [61], Aware App [74], Flowy [75] also showed a positive effect in reducing anxiety. Moreover, participants using the Loving-Kindness Meditation (LKM) app [77] reported a reduction in suicidal ideation. However, ABMT was not effective in improving depression and anxiety [78].

## Cognitive change

Thirty apps used cognitive change techniques like goal setting, adaptive challenges, structured problem-solving, motivational video playback, mood tracking, and interpretation bias training. Nineteen of these apps showed a positive effect in reducing depression, including the ibobbly app [46], the Cognitive Control (Project: EVO) [47], the Problem-Solving Therapy (iPST) app [47], Smartphone Positive Stimuli Response System (SPSRS) application [44,53], Feel Stress Free app [45], MoodPrism app [50], MoodMission app [51,57], BlueIce app [52], Headspace [48,49] and Smiling Mind [48], Mello App [54], Coping Camp app [55], Anchored app [58], Subliminal Priming with Supraliminal Reward Stimulation (SPSRS) [59], eQuoo app [60], MindLAMP app [61], FertiStrong app [62], CareMom app [64], MoodHacker app [66].

Out of 30, 19 apps had a positive impact on anxiety levels, including the Smartphone Positive Stimuli Response System (SPSRS) application [44], Feel Stress Free app [45], MoodPrism app [50], Interpretation Bias Evaluation app [67], MoodMission app [51], Insight Timer app [68], BlueIce app [52], Mello App [54], Coping Camp app [55], GAMA-AIMS [69], COVID

**Table 2. Description of all the apps identified in the review (n = 35).**

| Apps | Description | Function | CBT methods |
|---|---|---|---|
| Smartphone Positive Stimuli Response System [44,53] | It is a video playback app using YouTube API, featuring motivational words like "can," "good luck," and "enjoyable," displayed on-screen to boost self-confidence in young adults with stress depression, with a recommended usage of 70 minutes per week over 5 weeks | Cognitive change; Context engagement | Motivational words; Video playback. |
| Indoor exercise app [43] | It helps users with indoor physical exercises, helping individuals develop healthier associative patterns through structured physical activities. | Context engagement | Physical activity tracking |
| iBobbly [46] | It includes 3 self-assessment modules designed to help manage mental health. Module 1 employs context engagement techniques, guiding users to identify and manage their thoughts, feelings, and behaviors, including suicidal ideation. This module aims to reduce distress and enhance understanding of the relationship between experiences and mental health. Module 2 focuses on attention change, and Module 3 helps users identify personal values, set small, achievable goals, and create a personalized action plan. | Cognitive change; Context engagement; Attention change | Identifying Thoughts and Behaviors; Self-Assessments; Mindfulness; |
| MoodMission [51,57] | It engages users by helping them identify and report their current mood or anxiety levels. This initial step involves users describing their emotional state and selecting from a range of "Missions" based on their reported feelings. The app helps users become aware of their mood and begin the self-help process. | Cognitive change; Attention change; Context engagement | Behavioral activation activities; Mindfulness; Physical exercises |
| BlueIce app [52] | It facilitates maintaining mood diary and distress tolerance activities. The mood diary in the app allows users to record their emotional experiences and reflect on patterns or triggers associated with their self-harm behaviors to understand their emotional context and develop a more informed perspective on their mental health. | Cognitive change; Attention change; Context engagement | Thought-challenging exercises; Mindfulness; Toolbox of mood-lifting activities |
| Headspace [48,49,72,73] | It offers structured mindfulness meditation sessions engaging users through structured mindfulness practices focusing on focused breathing, body scans and improving sleep. | Cognitive change; Attention change; Context engagement | Reframing stress; Focused breathing exercises; Daily meditation practice |
| Smiling Mind app [48] | It provides age-specific mindfulness practices with thought-challenging exercises. | Cognitive change; Attention change; Context engagement | Mindful breathing, Body scans, Reframing negative thoughts; |
| Insight Timer [68] | It engages users with a wide variety of meditation practices such as guided breathing, guided imagery, body scanning, gratitude, and affirmations, allowing them to select sessions based on their current mood or specific needs and helps users address their emotional state and choose practices that fit their immediate context. | Cognitive change; Attention change; Context engagement | Gratitude practices and affirmations; Guided breathing and body scanning; Daily meditation routine |
| Therapeutic Evaluative Conditioning [76] | It provides a game-like environment that allows users to engage in tasks that help them reinterpret self-related stimuli and reduce self-injurious thoughts and behaviors, with each session lasting only 1–2 minutes. | Cognitive change; Context engagement | Interpretation bias training; Gamified challenges |
| The Feel Stress Free app [45] | It facilitates activities like calm breathing, mindfulness meditation, deep muscle relaxation, self-hypnosis, and mood tracking, guided by a friendly robot character, with a recommendation for weekly usage of at least 10 minutes. | Attention change; Cognitive change | Mindfulness; Mood tracking |
| Attention Bias Modification Training app [78] | It is a mobile gaming app designed to administer a single session of Attention Bias Modification Training (ABMT) to individuals showing higher levels of anxiety. This app engages the user in tasks that redirect attention away from threats. | Attention change | Attention bias modification training |
| Cognitive Control app [47] | It is a video game-based app that uses adaptive challenges to improve cognitive control abilities like working memory and attention for about 30 minutes a day, six days a week. | Cognitive change; | Cognitive control training; Structured; Daily health advice |
| Problem-Solving Therapy app [47] | It involves users setting personal goals and following a structured 7-step problem-solving process, which helps them create actionable plans for mood management. | Cognitive change; Context engagement | problem-solving; Goal setting; Daily health advice |
| MoodPrism [50] | It is a mood-tracking app. It prompts users for daily mood surveys, offering feedback and assessments over time to aid in understanding emotional states and mental wellbeing, with features including baseline and follow-up surveys for depression, anxiety, and mental health literacy. | Cognitive change | Mood tracking |
| The Interpretation Bias Evaluation app [67] | It helps individuals develop positive interpretations of unclear situations. Through a spelling completion task, participants engage with scenarios and incomplete phrases. The training, consisting of 50 trials per session, is conducted 3–5 times weekly for 5–10 minutes over 2.5 to 3 months. | Cognitive change | Interpretation bias training |

*(Continued)*

**Table 2.** (Continued)

| Apps | Description | Function | CBT methods |
|------|-------------|----------|-------------|
| Mental App [79] | It is designed for university students. It features self-monitoring of daily conditions and self-screening. It also provides advice to users on how to improve their physical and mental conditions according to the results of the daily record like "You should eat more", "You had enough time to sleep", "You need to exercise more", and user-recorded data like appetite, sleep, exercise, and mood. | Cognitive change | Mood tracking |
| MindSurf [80] | It sends metacognitive questions via text messages to users' phones at random times throughout the day, helping them to reframe their thoughts throughout the day. | Cognitive change | Thought challenging |
| Mello App [54] | It sends 3 daily notifications prompting brief check-ins to assess mood, repetitive negative thinking (RNT), activities, and location via a chat interface. Based on responses, the app recommends 1 of 12 cognitive behavioral therapy activities to complete in the moment. | Cognitive change; Context engagement; Attention change | Mindfulness; Problem-Solving; Thought Challenging; Self-Compassion |
| Coping Camp app [55] | It comprises 11 sequentially unlocked sessions focusing on stress education, skills training, and application. It features notifications, three locked assessments, and a monitored discussion board for peer support | Cognitive change; Context engagement; Attention change | Relaxation techniques; Behavioral skills like goal setting; Time management; Problem-solving |
| GAMA-AIMS [69] | It is an unguided digital self-help app, based on Beckian Cognitive Therapy, supports anxiety management through psychoeducational methods. It features sections for information about anxiety, therapy with 8 cognitive-behavioral modules, and a daily journal with tools like a mood tracker and e-diary. | Cognitive change; Attention change | Negative thought identification; Reframing; Relaxation |
| COVID Coach [70] | It is a self-management app designed to support mental well-being during the COVID-19 pandemic. It includes tools for stress management, educational content on coping and safety during the pandemic, mood tracking for symptoms like anxiety, and links to mental health and crisis resources. | Cognitive change; Context engagement; Attention change | Relaxation; Mindfulness; Mood tracking |
| Down Dog app [56] | It is a self-guided fitness app offering personalized exercise options, including bodyweight interval training, yoga, running, and barre. Users are prompted to complete four 20-minute sessions weekly, totaling 80 minutes, for 12 weeks, at home or any convenient location. | Context engagement; Attention change | Mindfulness; Physical activity; Yoga |
| Loving-Kindness Meditation app [77] | It is a short video app that delivers 5-minute animated guides for loving-kindness meditation, teaching users to practice compassionate blessings progressively, from themselves to all living beings. | Cognitive change; Context engagement; Attention change | Reframing negative thoughts; Guided meditation |
| Anchored app [58] | It is designed for individuals experiencing work-related stress. Its main feature is a 30-day intervention where users complete one daily 5–10 minute "challenge" incorporating behavioral activation, mindfulness, and CBT techniques via text, audio, images, and videos. | Reframing negative thoughts, Mindfulness, | Reframing negative thoughts, Mindfulness, |
| ImExposure app [71] | It guides users through imaginal exposure therapy for social anxiety by helping them create a personalized fear hierarchy, rating anxiety levels for social situations, and engaging in 13-minute audio-guided visualization exercises. Users reflect on their anxiety before, during, and after exercises and track progress while being encouraged to translate imagined scenarios into real-world exposure. | Cognitive restructuring; In vivo exposure; Exposure-based attentional focus | Cognitive restructuring; In vivo exposure; Exposure-based attentional focus |
| Subliminal Priming with Supraliminal Reward Stimulation [59] | It delivers a 10-minute video with repeated positive word stimulation. Subliminal words (e.g., "able," "good luck") appear briefly in the screen corners, and supraliminal positive words (e.g., "great," "fantastic") appear in the center every five seconds. Participants watch a basketball game video, selected to enhance mood and avoid inappropriate content, aiming to reduce depressive symptoms. | Reframing thoughts, Positive word priming | Reframing thoughts, Positive word priming |
| eQuoo app [60] | It is a gamified mental health tool for anxiety and depression prevention, combining CBT methods and positive psychology. Players customize avatars, learn psychological skills, and practice them through interactive stories and quizzes. Weekly level locks encourage real-life application, while gamification elements like rewards and quests enhance engagement. | Cognitive change; Attention change; Context engagement | Reframing negative thought; Focused tasks; Gamified challenges |
| MindLAMP app [61] | It is an open-source mental health app that combines digital phenotyping with mindfulness and CBT-based interventions. The app features daily and bi-weekly surveys, sent via push notifications. Users engage with mindfulness activities, guided meditation, and tailored interventions while receiving optional virtual support from digital navigators. | Cognitive change; Attention change; Context engagement | Reframing negative thought; Mindfulness; Guided meditations |

*(Continued)*

**Table 2.** (Continued)

| Apps | Description | Function | CBT methods |
|------|-------------|----------|-------------|
| FertiStrong app [62] | It is a self-guided app designed for men experiencing infertility and provides cognitive-behavioral coping strategies and relaxation techniques tailored to 50 common stress-inducing situations, including partner communication, pregnancy loss, and work stress. | Cognitive change; Context engagement | Relaxation exercises; Guiding strategies for real-life stress situations |
| Welzen app [63] | It offers a 7-day program repeated over 28 days, with daily 10-minute guided meditations focusing on stress recognition, breathing techniques, patience, relaxation, balancing external demands with inner peace, and cultivating self-compassion through loving-kindness meditation. | Attention change; Context engagement | Mindfulness; Meditation; Relaxation |
| CareMom app [64] | It delivers short videos (2–4 minutes) on topics like cognitive distortions, human emotions, and strategies for challenging negative thoughts, followed by quiz questions to reinforce learning. The challenges are automatically released daily, ensuring consistent engagement, and users can complete missed challenges later. Mood management tools are also included to help users track and reflect on their emotional well-being throughout the program. | Cognitive change | Reframing cognitive distortions |
| We'll App [65] | It is a mindfulness practices app included: body scanning, meditation, mindful walking, postpartum health education, and social support through a network of family, friends, and other puerperae. | Attention change; Context engagement | Mindfulness practice; Social support network and interactive features |
| Aware App [74] | It offers a guided mindfulness meditation program based on the Kabat-Zinn mindfulness meditation framework. It includes foundational mindfulness techniques designed to enhance users' mindfulness skills. | Cognitive change; Attention change; Context engagement | Mindfulness, Meditation, Awareness of thoughts and feelings |
| Flowy [75] | It is a mobile health minigame app using diaphragmatic breathing exercises to reduce anxiety. Users control minigames by syncing their breathing with a visual indicator, learning relaxation techniques through guided tutorials. Progress depends on breathing correctly, promoting calmness naturally. | Cognitive change; Attention change; Context engagement | Reframing anxiety; Guided and Integrates breathing |
| MoodHacker app [66] | It is a 6-week mobile web app, which provides daily mood and activity tracking, positive behavior engagement, and mindfulness practices. Users receive structured content via emails, in-app messages, articles, and videos include goal setting, journaling, and a tracker to monitor progress, promoting self-management and resilience. | Cognitive change; Attention change; Context engagement | Reframing negative thought; Mindfulness techniques; Social and physical tasks |

Coach [70], Anchored app [58], ImExposure (IE) app [71], Headspace [72,73], eQuoo app [60], MindLAMP app [61], FertiStrong app [62], Aware App [74], and Flowy [75]. The Blue-Ice app [52] and the Loving-Kindness Meditation (LKM) app [77] reduced self-harm symptoms and suicide ideation, respectively. Similarly, the TEC app [76] demonstrated a significant decrease in self-cutting episodes, suicide plans, and suicidal behaviors. In contrast, the Mental App [79] and MindSurf app [80] did not significantly improve mental health.

As detailed in Table 3, we identified 71 unique survey instruments used by different studies in the review. Among these, the Patient Health Questionnaire-9 (PHQ-9; n = 10), Generalized Anxiety Disorder-7 (GAD-7; n = 16), and Depression Anxiety Stress Scales-21 (DASS-21; n = 5) were the most frequently used.

## Discussion

Our review investigates the impact of mHealth apps in mitigating mental health issues, focusing on depression, anxiety, and suicidal behaviors. Our findings show that context engagement and cognitive change techniques are the most effective CBT methods for mental health apps.

Our review also highlights a gap in the efficacy of mHealth apps for managing more complex mental health conditions, such as suicidal behaviors, where evidence remains scant and less definitive. Only a few apps specifically target suicidal ideation or behaviors. For example, among the apps reviewed, BlueIce, the Loving-Kindness Meditation (LKM) app, and

**Table 3. Survey instruments identified in the review.**

| Study | Survey instrument |
|---|---|
| [53] | CES-D, GSES, GHQ-12, LSAS, IL-6 Levels |
| [44] | CES-D, K-6, GAD-7 |
| [79] | CES-D, GHQ-12 |
| [45] | HADS-A, HADS-D |
| [76] | SITBI, ERS, BSI, IDB, AMP |
| [50] | PHQ-9, WEMWBS, ESAS-R, MHLQ, CSES, Mobile Application Rating Scale, SDS. |
| [80] | DASS-21 |
| [67] | DASS-21, IAS |
| [43] | DASS-21, IAS, UWES, WHO, OSQ |
| [78] | STAI, BDI-II, POMS, Mini-MAC |
| [46] | DSI-SS, PHQ-9, K10, BIS-11 |
| [47] | PHQ-9, SDS, GAD-7, IMPACT, Assessment of Mania and Psychosis, AUDIT-C |
| [51] | PHQ-9, GAD-7, WEMWBS, ESAS-R, CSES, MHLQ, App Engagement Scale |
| [52] | GAD-2, PHQ-2, ALSPAC |
| [48] | CES-D, PSS, BRS, FS, CAT, CAMS-R |
| [68] | GAD-7, World Health Organization-Five Well-Being Index (WHO-5) |
| [49] | PHQ-9 |
| [54] | PHQ-8, GAD-7, PTQ |
| [55] | DASS-21, ORS-4, CISS-SFC, PSS-10 |
| [69] | GAD-7, TMAS |
| [70] | CES-D-10, STAI -S, ASDS, PSS-4, WHO-5 |
| [56] | CES-D-10, MBI |
| [77] | MAAS, PCQ, RSCS-C, BSSI |
| [57] | PHQ-9, GAD-7. |
| [58] | PHQ-9, GAD-7, PSS, WHO-5, BRS, AQoL-4D, CBI-WBI |
| [71] | SIAS, SPDQ, GSES, IPQ, QUMI, PIQ |
| [59] | POMS 2-A, STAI-S |
| [72] | PSS, GAD-7, MBI |
| [60] | RRM, GAD-7, PHQ-8 |
| [61] | PHQ-9, GAD-7 |
| [62] | HADS, FPI |
| [63] | PSS, GAD-7, BDI-21, FFMQ-SF |
| [64] | EPDS, GAD-7 |
| [73] | GAD-7, PHQ-9, ProQOL |
| [65] | MAAS, MSPSS, PMPS, EPDS |
| [74] | DASS-21 |
| [75] | GAD-7, OASIS, ASI-3, PDSS-SR |
| [66] | PHQ-9, BADS, ATQ-R |

CES-D = Center for Epidemiologic Studies Depression Scale; IL-6 = Salivary Interleukin-6; GSES = General Self-Efficacy Scale; GHQ-12 = 12-Item General Health Questionnaire; LSAS = Liebowitz Social Anxiety Scale; K-6 = Kessler Screening Scale for Psychological Distress; GAD-7 = Generalized Anxiety Disorder-7; HADS-A = HADS-Anxiety Subscale; HADS-D = HADS-Depression Subscale; SITBI = Self-injurious Thoughts and Behaviors Interview; ERS = Emotion Reactivity Scale; BSI = Brief Symptom Inventory; IDB = Index of Dysregulated Behaviors; AMP = Affect Misattribution Procedure; PHQ-9 = Patient Health Questionnaire 9-Item; WEM-WBS = Warwick-Edinburgh Mental Wellbeing Scale; ESAS-R = Emotional Self-Awareness Scale-Revised; MHLQ = Mental Health Literacy Questionnaire; CSES = Coping Self-Efficacy Scale; SDS = Social Desirability Scale; DASS-21 = Depression Anxiety and Stress Scale 21; IAS = Interaction Anxiousness Scale; UWES = Utrecht Work Engagement Scale; OSQ = Occupational Stress Questionnaire; STAI = State-Trait Anxiety Inventory; BDI-II = Beck Depression Inventory II; POMS = Profile of Mood States; Mini-MAC = Mini-Mental Adjustment to Cancer; DSI-SS = Depressive Symptom Inventory Suicidality Subscale; K10 = Kessler Psychological Distress Scale; BIS-11 = Barratt Impulsivity Scale; IMPACT = Improving Mood-Promoting Access to Collaborative Treatment; Assessment of Mania and Psychosis; AUDIT-C = Alcohol Use Disorders Identification Test; ALSPAC = Avon Longitudinal Study of Parents and Children Adapted Self-harm Questions; PSS = Perceived Stress Scale;

*(Continued)*

**Table 3.** (Continued)

BRS = Brief Resilience Scale; FS = Flourishing Scale; CAT = College Adjustment Test; CAMS-R = Cognitive Affective Mindfulness Scale–Revised; WHO-5 = World Health Organization-Five Well-Being Index; PTQ = Perseverative Thinking Questionnaire; ORS-4 = Outcome Rating Scale; CISS-SFC = Coping Inventory for Stressful Situations; TMAS = Taylor Manifest Anxiety Scale; ASDS = Acute Stress Disorder Scale; MBI = Maslach Burnout Inventory; MAAS = Mindfulness Attention Awareness Scale; PCQ = Psychological Capital Questionnaire; RSCS-C = Self-Compassion Scale; BSSI = Beck Scale of Suicidal Ideation; BRS = Brief Resilience Scale; AQoL-4D = Assessment of Quality of Life-4D; CBI-WBI = Copenhagen Burnout Inventory-Work Burnout Index; SIAS = Social Interaction Anxiety Scale; SPDQ = Social Phobia Diagnostic Questionnaire; IPQ = Igroup Presence Questionnaire; QUMI = Questionnaire Upon Mental Imagery; PEIQ = Post-exercise Imagery Questions; RRM = Rugged Resilience Measure; HADS = Hospital Anxiety and Depression Scale; FPI = Fertility Problem Inventory; FFMQ-SF = Five Facet Mindfulness Questionnaire; EPDS = Edinburgh Postnatal Depression Scale; ProQOL = Professional Quality of Life scale; MSPSS = Multidimensional Perceived Social Support Scale; PMPS = Perceived Maternal Parental Self-Efficacy; OASIS = Overall Anxiety Severity and Impairment Scale; ASI-3 = Anxiety Sensitivity Index-3; PDSS-SR = Panic Disorder Severity Scale-Self Report; BADS = Behavioral Activation for Depression Scale; ATQ-R = Automatic Thoughts Questionnaire-Revised.

the Therapeutic Evaluative Conditioning (TEC) app include features designed to help users manage self-harming and suicidal thoughts. These apps employ distress tolerance techniques, mood tracking, and crisis management features to address suicidal behaviors, but the evidence supporting their efficacy remains preliminary. This mirrors concern in the broader literature about the challenges of addressing high-risk mental health conditions through app-based interventions alone.

Our findings underscore the potential of mHealth applications in providing adequate mental health interventions. For populations in remote or underserved regions, where traditional mental health services are scarce or non-existent, mHealth apps can offer a viable avenue for receiving support [81]. Apps that provide self-monitoring and self-help strategies enable users to begin addressing their mental health issues in the early stages, potentially preventing the escalation of symptoms [82]. This early intervention approach can improve individual outcomes while reducing the overall burden on healthcare systems. Integrating mHealth applications into traditional healthcare systems presents a promising avenue as well. By supplementing face-to-face therapy with app-based interventions, healthcare providers can offer continuous support and monitoring, extending the therapeutic engagement beyond the clinical setting. For example, apps facilitating cognitive-behavioral therapy activities or mood tracking can augment therapeutic strategies employed by mental health professionals, creating an integrated care model that capitalizes on the strengths of both digital and traditional methods.

Evaluating mental health apps often hinges on their ability to demonstrate tangible improvements in mental health outcomes. In our study, out of the 35 apps reviewed, three apps, namely ABMT, Mental app, and MindSurf app, did not significantly improve mental health outcomes [78–80]. A common characteristic of these studies was their smaller sample sizes, ranging from 20 to 60 participants. In contrast, studies with larger sample sizes, such as the MoodPrism app (N = 168) [50], MoodMission app (N = 617) [51], Headspace and Smiling Mind apps (N = 208) [48], the iPST and Project: EVO apps (N = 626) [47], and Feel Stress Free app (N = 198) [45], demonstrated improvements in mental health outcomes. This raises important considerations regarding the sample size and its influence on the ability to measure the true impact of app interventions. The importance of sample size in research cannot be overstated. The lack of observed improvement in mental health outcomes in the ABMT, Mental app, and MindSurf app studies could be partially attributed to insufficient sample sizes, which may not provide a robust test of the apps' efficacy.

Additionally, out of 35 apps, three apps were investigated in multiple studies: the Smartphone Positive Stimuli Response System (SPSRS) [44,53], the MoodMission app [51,57], and the Headspace app [48,49,72,73] with different populations, study duration, and outcomes. For example, a study by Bakker et al.[51] examined the MoodMission app's impact on a broader population (N = 617), ranging from adolescents to older adults, emphasizing

self-guided engagement over 30 days. The findings revealed that the app effectively reduced symptoms of depression and anxiety, particularly for individuals with moderate baseline symptoms, showcasing its utility in promoting coping self-efficacy among a general population. In contrast, another study by Tan et al., [57] with a smaller clinical population of psychiatric outpatients (N = 48), evaluated the MoodMission app as an adjunct to treatment over four weeks. While the app significantly reduced depressive symptoms in this setting, it did not yield significant improvements in anxiety symptoms, suggesting that its effectiveness may depend on the target population and treatment context. Similarly, Headspace was examined in four studies differing in duration, sample size, and outcomes. One study with university students (N = 208) over 30 days reported reductions in depressive symptoms [48]. Another study with college students (N = 72) over 14 days found significant decreases in depressive symptoms [49]. An 8-week intervention with orthopedic surgery residents (N = 24) showed reductions in anxiety with minimal app use (8 minutes per day, 2 days per week) [72]. However, a smaller study with surgical residents and faculty (N = 19) over 14 days reported anxiety reduction but no significant improvements in depressive symptoms [73]. These findings illustrate how differences in intervention duration, population size, and context influence reported outcomes and highlight the importance of tailoring apps used to specific settings and populations.

While our review sheds light on the potential of mHealth apps, a closer examination of the current body of research reveals a critical gap: most of these studies are early-phase trials or pilot studies. While these studies provide valuable insights into their preliminary impact, they lack the rigorous clinical validation found in later-phase trials. This underscores the need to advance mHealth research to phases 2 and 3 trials to ensure clinical validation and facilitate the potential adoption of such apps into mainstream mental health care. For mHealth apps, transitioning into these later phases is imperative to validate their therapeutic value against standardized clinical benchmarks. Such rigorous testing ensures that apps can genuinely benefit users in real-world settings beyond the controlled environments of research studies. Moreover, phase 2 and 3 trials incorporate larger, more diverse participant groups, enhancing the generalizability of findings [83]. Only through such comprehensive evaluation can we identify which apps are genuinely efficacious across different populations, mental health disorders, and severity levels.

One of the limitations of this review study is the exclusion of non-English publications, which may have led to a language bias and the omission of relevant studies from non-English-speaking countries.

## Conclusion

In conclusion, mHealth apps are promising for addressing the global mental health crisis, offering scalable, accessible interventions. However, the current evidence base highlights the need for more robust, long-term studies to understand their efficacy better and develop guidelines for their integration into mainstream mental health care. As we move forward, mHealth interventions must be designed and evaluated, emphasizing evidence-based practices, user engagement, and inclusivity to maximize their impact on mental health outcomes worldwide.

## Supporting information

**S1 File. PRISMA checklist.**
(DOCX)

**S2 File. Search query.**
(DOCX)

**S3 File. Mixed methods appraisal tool.**
(DOCX)

**S4 File. Article screening prosses and exclusion reasoning.**
(DOCX)

## Author contributions

**Conceptualization:** Avishek Choudhury.

**Data curation:** Yeganeh Shahsavar, Avishek Choudhury.

**Formal analysis:** Yeganeh Shahsavar, Avishek Choudhury.

**Funding acquisition:** Avishek Choudhury.

**Investigation:** Avishek Choudhury.

**Methodology:** Yeganeh Shahsavar, Avishek Choudhury.

**Resources:** Avishek Choudhury.

**Software:** Avishek Choudhury.

**Supervision:** Avishek Choudhury.

**Validation:** Avishek Choudhury.

**Visualization:** Yeganeh Shahsavar.

**Writing – original draft:** Yeganeh Shahsavar, Avishek Choudhury.

**Writing – review & editing:** Avishek Choudhury.

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
