## [Decision Letter · Decision Letter 0]

31 Oct 2024

Dear Dr. Choudhury,

Thank you for submitting your manuscript to PLOS ONE. After careful consideration, we feel that it has merit but does not fully meet PLOS ONE’s publication criteria as it currently stands. Therefore, we invite you to submit a revised version of the manuscript that addresses the points raised during the review process.

As you will notice, I was only able to secure one reviewer for this manuscript. I conducted a review as well and agree with their helpful and clear comments. If you choose to revise and resubmit this manuscript, please address all of their concerns. If you resubmit, I might secure an additional reviewer.

We look forward to receiving your revised manuscript.

Kind regards,

Ethan Moitra

Academic Editor

PLOS ONE

Journal Requirements:

2. Please identify your study as "scoping review" OR "systematic review" OR "systematic review and meta-analysis" in the title of your manuscript.

3. Thank you for stating the following financial disclosure: “This work was supported by West Virginia University Internal Seed Grant (#3086)”

4. We note that your Data Availability Statement is currently as follows: “All relevant data are within the manuscript and in Supporting Information files.”

Please confirm at this time whether or not your submission contains all raw data required to replicate the results of your study. Authors must share the “minimal data set” for their submission. PLOS defines the minimal data set to consist of the data required to replicate all study findings reported in the article, as well as related metadata and methods (https://journals.plos.org/plosone/s/data-availability#loc-minimal-data-set-definition). For example, authors should submit the following data: - The values behind the means, standard deviations and other measures reported; - The values used to build graphs; - The points extracted from images for analysis. Authors do not need to submit their entire data set if only a portion of the data was used in the reported study. If your submission does not contain these data, please either upload them as Supporting Information files or deposit them to a stable, public repository and provide us with the relevant URLs, DOIs, or accession numbers. For a list of recommended repositories, please see https://journals.plos.org/plosone/s/recommended-repositories. If there are ethical or legal restrictions on sharing a de-identified data set, please explain them in detail (e.g., data contain potentially sensitive information, data are owned by a third-party organization, etc.) and who has imposed them (e.g., an ethics committee). Please also provide contact information for a data access committee, ethics committee, or other institutional body to which data requests may be sent. If data are owned by a third party, please indicate how others may request data access.

5. Please amend either the title on the online submission form (via Edit Submission) or the title in the manuscript so that they are identical.

6. As required by our policy on Data Availability, please ensure your manuscript or supplementary information includes the following: A numbered table of all studies identified in the literature search, including those that were excluded from the analyses. For every excluded study, the table should list the reason(s) for exclusion. If any of the included studies are unpublished, include a link (URL) to the primary source or detailed information about how the content can be accessed. A table of all data extracted from the primary research sources for the systematic review and/or meta-analysis. The table must include the following information for each study: Name of data extractors and date of data extraction Confirmation that the study was eligible to be included in the review. All data extracted from each study for the reported systematic review and/or meta-analysis that would be needed to replicate your analyses. If data or supporting information were obtained from another source (e.g. correspondence with the author of the original research article), please provide the source of data and dates on which the data/information were obtained by your research group. If applicable for your analysis, a table showing the completed risk of bias and quality/certainty assessments for each study or outcome. Please ensure this is provided for each domain or parameter assessed. For example, if you used the Cochrane risk-of-bias tool for randomized trials, provide answers to each of the signalling questions for each study. If you used GRADE to assess certainty of evidence, provide judgements about each of the quality of evidence factor. This should be provided for each outcome. An explanation of how missing data were handled. This information can be included in the main text, supplementary information, or relevant data repository. Please note that providing these underlying data is a requirement for publication in this journal, and if these data are not provided your manuscript might be rejected.

Reviewers' comments:

Reviewer's Responses to Questions

**Comments to the Author**

1. Is the manuscript technically sound, and do the data support the conclusions?

Reviewer #1: Partly

2. Has the statistical analysis been performed appropriately and rigorously?

Reviewer #1: N/A

3. Have the authors made all data underlying the findings in their manuscript fully available?

Reviewer #1: Yes

4. Is the manuscript presented in an intelligible fashion and written in standard English?

Reviewer #1: Yes

Reviewer #1: This paper provides information about the effectiveness of evidence-based health apps on user mental health outcomes. The article provides information with high clinical relevance in the field of mental health interventions. However, some aspects must be improved to facilitate understanding. In this sense, I would like to add some comments/suggestions to this paper:

• It is recommended to include the protocol registration number (line 103).

• Why have authors limited the objective of the study only to these mental health problems (depression, anxiety, bipolar disorder, psychosis, and suicidal behaviors)?

• Some search terms (depression, anxiety, suicide, mobile, app, and smartphone) do not match the aim of the study (depression, anxiety, bipolar disorder, psychosis, and suicidal behaviors). Bipolar disorder and psychosis are not included as search terms.

• It is recommended to provide more information about the methodology for selecting articles (e.g., if it was done by title and abstract; if, in the event of not reaching an agreement, they included third parties in the decisión; what specific software was used to remove duplicates and for screening abstract).

• Were all the apps based on the cognitive behavioral approach? Did any applications include functions from other approaches? In order to better understand the results, it is recommended to include a table specifying the components of the different apps (e.g., mindfulness, physical activity tracking) for each CBT-function.

• It is recommended to specify the number of articles deleted by title and the number of articles deleted by abstract separately.

• Was any quality assessment carried out on the articles used? For example, through the Effective Public Health Practice Project Quality Assessment Tool (EPHPP)

• In Table 1 it is recommended to indicate the standard deviation of the age of the participants.

• It is recommended to review and unify the uppercase and lowercase letters used in the instruments column of table 1

• On lines 176-177 it is recommended to indicate the number of studies.

• The improvement in mental health outcomes is mentioned in the discussion section. However, few applications address suicidal behaviors. It is recommended that this section be redone, specifying mental health information obtained in more detail.

• Table 3 does not indicate whether the results obtained were short or long term.

**Do you want your identity to be public for this peer review?** For information about this choice, including consent withdrawal, please see our Privacy Policy

Reviewer #1: No

---

## [Author Response · Author response to Decision Letter 1]

22 Jan 2025

Response to Editor

Dear editor. Thank you for giving us the chance to improve the manuscript. Given the delay, we decided to redo the entire review and update our results. We have addressed all the comments and more. We were asked to include a list of all papers screened, with their DOI/Link and reason for exclusion (which I have never seen in my life before). All this information has been provided in S4 – however, I am unsure what value it will add to our work.

Response to Reviewers

Reviewers

Reviewer #1: This paper provides information about the effectiveness of evidence-based health apps on user mental health outcomes. The article provides information with high clinical relevance in the field of mental health interventions. However, some aspects must be improved to facilitate understanding. In this sense, I would like to add some comments/suggestions to this paper:

• It is recommended to include the protocol registration number (line 103).

Response: We added the OSF protocol citation.

“The detailed protocol (osf.io/x6m7u) is registered at the Open Science Framework [39].”

• Why have authors limit the objective of the study only to these mental health problems (depression, anxiety, bipolar disorder, psychosis, and suicidal behaviors)?

Response: We agree. However, reviews on other mental health disorder already exist and for some disorders, there are no apps for us to evaluated. We have clarified this in the manuscript.

“Few existing review in this field have focused on feasibility of various apps designed for agoraphobia, eating disorders, post-traumatic stress disorders (PTSD), substance use disorders, sleep disorders [34, 35]. However, there is lack of evidence reviewing effectiveness of mobile apps in improving mental health outcomes [36, 37].”

• Some search terms (depression, anxiety, suicide, mobile, app, and smartphone) do not match the aim of the study (depression, anxiety, bipolar disorder, psychosis, and suicidal behaviors). Bipolar disorder and psychosis are not included as search terms.

Response: Thank you for pointing this out. We started with those keywords, but there are studies involving an app that matches our inclusion criteria. We then revised our search term in the manuscript to avoid further discrepancies.

“Search query consisted of the following ((Depression OR anxiety OR suicid*) AND (Mobile OR app OR smartphone)) NOT (sleep OR alcohol OR drugs OR addiction OR tobacco) (see S2)”.

• It is recommended to provide more information about the methodology for selecting articles (e.g., if it was done by title and abstract; if, in the event of not reaching an agreement, they included third parties in the decision; what specific software was used to remove duplicates and for screening abstracts).

Response: We have now updated this section to specify that Excel sheets were used for duplicate removal, decisions on conflicts were resolved through discussion between the two authors without the involvement of third parties and how titles and abstracts were screened.

“The methodology for selecting studies involved a multi-step process. Two authors independently selected the studies using the inclusion and exclusion criteria. Conflicts were then resolved with discussion, without the involvement of third parties. First, duplicates were identified and removed using Excel sheets created from the database exports. Titles of potentially relevant studies were then screened manually to eliminate irrelevant articles. This was followed by a review of abstracts to exclude studies that did not use mobile phone technology, did not focus on mental health apps, lacked emphasis on treatment or mental health impact, or addressed unrelated topics like mobile or technology addiction. Finally, full-text studies were evaluated against the inclusion and exclusion criteria”.

• Were all the apps based on the cognitive behavioral approach? Did any applications include functions from other approaches? To better understand the results, it is recommended to include a table specifying the components of the different apps (e.g., mindfulness, physical activity tracking) for each CBT function.

Response: All the apps included in this study were based on the cognitive behavioral approach, as their core components (e.g., cognitive change, attention change, and context engagement) align with CBT principles. While some functions, such as distress tolerance (BlueIce app) and self-soothing (iBobbly app), share similarities with techniques used in Dialectical Behavior Therapy (DBT), these were integrated within a CBT framework. Additionally, general wellness strategies, such as physical activity tracking and gamified challenges, complement CBT techniques by enhancing behavioral engagement. To address this comment, we have added an extra column to Table III that specifies the components of each app (e.g., mindfulness, physical activity tracking) and their alignment with CBT functions, ensuring comprehensive understanding.

• It is recommended to specify the number of articles deleted by title and the number of articles deleted by abstract separately.

Response: Thank you for your suggestion. We have included the specific numbers of articles deleted at each stage (by title and by abstract) in the manuscript. Also see S4.

• Was any quality assessment carried out on the articles used? For example, through the Effective Public Health Practice Project Quality Assessment Tool (EPHPP)

Response: Yes, we conducted a quality assessment of the included articles using the Mixed Methods Appraisal Tool (MMAT) instead of EPHPP.

Quality Assessment and Risk of Bias:

“We conducted a quality assessment of the papers following the Mixed Methods Appraisal Tool (MMAT) (see S3) [42]. The MMAT assesses the quality of qualitative, quantitative, and mixed methods studies. It focuses on methodological criteria and includes the nature of the study (randomized or nonrandomized clinical trial) across the performance bias, detection bias, attrition bias, and selection bias. No articles were excluded”.

• In Table 1 it is recommended to indicate the standard deviation of the age of the participants.

Response: We reviewed and updated Table 1 to include the mean and standard deviation (SD) of participant ages for studies where these values were explicitly reported in the original articles. For other studies that only provided age ranges, we were unable to calculate or include the SD due to the lack of detailed data.

• It is recommended to review and unify the uppercase and lowercase letters used in the instrument’s column of table 1

Response: Thank you for the suggestion. We have reviewed and updated the "Survey Instrument" column in Table 1 to ensure consistency in the use of uppercase and lowercase letters. All survey names now follow a standardized Title Case format with properly formatted acronyms in parentheses.

• On lines 176-177 it is recommended to indicate the number of studies.

Response: We have addressed this by including the number of studies corresponding to each approach in the manuscript.

• The improvement in mental health outcomes is mentioned in the discussion section. However, few applications address suicidal behaviors. It is recommended that this section be redone, specifying mental health information obtained in more detail.

Response: Thank you for your valuable feedback. We have revised the discussion section and provided more detailed information, particularly regarding suicidal behaviors.

Discussion:

“Our review also highlights a gap in the efficacy of mHealth apps for managing more complex mental health conditions such as suicidal behaviors, where evidence remains scant and less definitive. Only a few apps specifically target suicidal ideation or behaviors. For example, among the apps reviewed, BlueIce and TEC app include features designed to help users manage self-harming and suicidal thoughts. These apps employ distress tolerance techniques, mood tracking, and crisis management features to address suicidal behaviors, but the evidence supporting their efficacy remains preliminary. This mirror concerns in the broader literature about the challenges of addressing high-risk mental health conditions through app-based interventions alone.

• Table 3 does not indicate whether the results obtained were short or long term.

Response: We have added a new column, "Study duration," to Table 3 to specify the duration of each study (e.g., 3 weeks, 6 weeks).

---

## [Decision Letter · Decision Letter 1]

12 Feb 2025

Effectiveness of Evidence Based Mental Health Apps on User Health Outcome: A Systematic Literature Review

PONE-D-24-30773R1

Dear Dr. Choudhury,

We’re pleased to inform you that your manuscript has been judged scientifically suitable for publication and will be formally accepted for publication once it meets all outstanding technical requirements.

Kind regards,

Ethan Moitra

Academic Editor

PLOS ONE

Additional Editor Comments (optional):

Reviewers' comments:

Reviewer's Responses to Questions

**Comments to the Author**

Reviewer #1: All comments have been addressed

2. Is the manuscript technically sound, and do the data support the conclusions?

Reviewer #1: Yes

3. Has the statistical analysis been performed appropriately and rigorously?

Reviewer #1: Yes

4. Have the authors made all data underlying the findings in their manuscript fully available?

Reviewer #1: Yes

5. Is the manuscript presented in an intelligible fashion and written in standard English?

Reviewer #1: Yes

Reviewer #1: Thanks to the authors for carrying out a thorough review based on the indicated suggestions/comments.

**Do you want your identity to be public for this peer review?** For information about this choice, including consent withdrawal, please see our Privacy Policy

Reviewer #1: No

---

## [Editor Report · Acceptance letter]

PONE-D-24-30773R1

PLOS ONE

Dear Dr. Choudhury,

I'm pleased to inform you that your manuscript has been deemed suitable for publication in PLOS ONE. Congratulations! Your manuscript is now being handed over to our production team.

Kind regards,

on behalf of

Dr. Ethan Moitra

Academic Editor

PLOS ONE